# Bile Leak: Is There Optimal Timing for Endoscopy?

**DOI:** 10.3390/medicina61122108

**Published:** 2025-11-27

**Authors:** Theodoros A. Voulgaris, Ioannis S. Papanikolaou, Dimitrios I. Ziogas, George Tribonias, Aliki Stamou, Aspasia Louta, Konstantinos Iliakopoulos, Ioannis A. Vezakis, Andreas Polydorou, Antonios Vezakis

**Affiliations:** 1Department of Endoscopy, Second Academic Surgical Unit, National and Kapodistrian University of Athens, Aretaieion Hospital, 11528 Athens, Greece; thvoulgaris87@gmail.com (T.A.V.); alouta2208@gmail.com (A.L.); kiliakopoulos1@gmail.com (K.I.); apolyd@med.uoa.gr (A.P.); avezakis@hotmail.com (A.V.); 2Hepatogastroenterology Unit, Second Department of Internal Medicine-Propaedeutic, Medical School, National and Kapodistrian University of Athens, 12462 Athens, Greece; aliceastam@gmail.com; 3Department of Gastroenterology, Athens Naval Hospital, 11521 Athens, Greece; dimiziog95@gmail.com; 4Department of Gastroenterology, Red Cross Hospital, 11526 Athens, Greece; g.tribonias@gmail.com; 5Biomedical Engineering Laboratory, School of Electrical and Computer Engineering, National Technical University of Athens, Zografou Polytechnic Campus, 15772 Athens, Greece; ivezakis@biomed.ntua.gr

**Keywords:** endoscopic retrograde cholangiopancreatography, bile leak, stent, healing rate, timing

## Abstract

*Background and Objectives*: Bile leak is a common complication after hepatopancreatobiliary surgery, requiring timely management to prevent life-threatening outcomes. Endoscopic retrograde cholangiopancreatography (ERCP) is essential in treatment, but large data concerning optimal timing and technique selection are unavailable. This study evaluates whether the timing of ERCP influences healing and if different bile duct injuries affect outcomes. *Materials and Methods*: Data from a prospectively maintained database over 25 years (2001–2025) included 176 patients (M/F: 91/85, mean age 62) undergoing ERCP for bile leaks. *Results*: Most leaks followed cholecystectomy (n = 143, 81.5%). The median time from leak to ERCP was 7 days. Ten patients (5.7%) had complete common bile duct (CBD) transection—considered major leaks—requiring surgery. Among the 166 minor leaks, the cystic duct stump (40.1%) was the most common injury site, followed by the CBD (24.1%) and the gallbladder bed (15.4%). Healing occurred in 90.6%. Stent placement improved healing rates (93.9% vs. 75.9%, *p* = 0.007), with no difference between pig-tail and (Amsterdam) straight plastic stents (90% vs. 96%, *p* = 0.267). Retained CBD stones or CBD strictures did not affect outcomes. Leaks from the cystic duct stump had a 96.9% resolution rate, whereas gallbladder bed leaks healed in 88%. The median healing time was 2 days, unaffected by stent placement or ES alone (*p* = 0.842), but later ERCP correlated with longer healing (RR: 0.362, *p* < 0.001). Following a right aberrant bile leak, the time for healing was longer than in leaks from other sites. *Conclusions*: ERCP with stenting remains the first-line approach for minor bile leaks. Early ERCP accelerates healing, emphasizing the importance of prompt intervention.

## 1. Introduction

Bile leak is defined as the escape or leakage of bile from any part of the biliary system and is among the most common complications of hepatopancreatobiliary (HPB) surgery, particularly cholecystectomy. It can also result from liver trauma [1]. Postoperative bile leaks typically manifest as bile flow into drain reservoirs, but it may also present with fever, abdominal pain, (bilious ascites) bilomas, with or without jaundice, and bile leakages from incisions [2]. This condition necessitates early recognition and proper treatment due to the risk of life-threatening complications such as biliary peritonitis, sepsis, and long-term biliary fistula formation. Additionally, bile leaks prolong hospitalization, increase healthcare costs, and significantly impact morbidity and mortality [2,3,4,5].

Currently, endoscopic retrograde cholangiopancreatography (ERCP) plays a pivotal role in managing bile leaks. Although surgical repair remains the main treatment for major duct disruptions involving the complete loss of common and/or hepatic duct continuity, ERCP is considered the gold-standard initial intervention for most other cases [6]. While magnetic resonance cholangiopancreatography (MRCP) offers excellent diagnostic accuracy in identifying the leak site, it is generally not used as the primary modality because ERCP can be used to both diagnose and treat the underlying injury [6,7,8,9]. The therapeutic efficacy of ERCP relies on decreasing the biliary pressure by removing the transpapillary pressure gradient, thereby facilitating bile flow into the duodenum and promoting the healing of ruptured bile ducts [10].

Several studies have attempted to determine the optimal timing for ERCP in patients with bile leaks, as well as whether specific ERCP techniques and modalities can enhance healing rates. Data on the ideal timing are scarce. Existing research suggests that early ERCP (within the first three days) does not necessarily increase healing rates [11,12]. Regarding techniques, evidence now supports the notion that stent placement, with or without sphincterotomy, improves healing outcomes. Early studies indicated that sphincterotomy alone might suffice in low-grade leaks without additional risk factors such as biliary strictures, coagulopathy, or severe sepsis [13,14,15]. It is understood that stent placement reduces the biliary pressure and seals the leak point. Consequently, the latest European Society of Gastrointestinal Endoscopy (ESGE) guidelines recommend transpapillary stenting as the optimal method for managing bile leaks [16].

This study aims to evaluate the efficacy and outcomes of the endoscopic management of bile leaks of various etiologies. In particular, it seeks to assess whether timing influences healing and whether different bile duct injuries affect post-ERCP healing rates, as well as to confirm, on a large scale, the superiority of stenting over sphincterotomy alone in bile leak resolution.

## 2. Materials and Methods

### 2.1. Study Design

This was a real-world, retrospective study of consecutive ERCP procedures performed at two high-volume HPB centers in Athens, Greece: Tzaneio General Hospital and Aretaeion Hospital. Procedures were conducted by a senior endoscopist (AV) or supervised trainees under AV’s oversight. The study period spanned 25 years (January 2001–July 2025). Of 5983 patients who underwent ERCP during this period, 176 met the inclusion criteria after excluding those treated for indications other than a bile leak. Patients were referred for ERCP according to hospital policy if the postoperative drain output exceeded 500 cc on the first day, 300 cc after three days, or 100 cc after seven days, or if intra-abdominal fluid collections were documented via imaging.

### 2.2. Data Collection

All information was recorded prospectively in a predefined form, standardized for all patients throughout the 25-year study period. Epidemiological, clinical, laboratory, radiological, and therapeutic data, as well as post-ERCP follow-up information, were collected from a maintained database in 2025 and analyzed retrospectively (see Figure 1 for the study flowchart). Missing data were handled through pairwise deletion, utilizing all available data points for each analysis, without imputation.

### 2.3. ERCP Procedure

Cannulation of the common bile duct (CBD) was attempted with a sphincterotome and guidewire. No contrast was injected into the pancreatic duct unless necessary. A needle-knife precut was performed after 5–10 min of failed cannulation or multiple unsuccessful attempts (>5 attempts). Once cannulation was achieved, a sphincterotomy was performed, and stenting decisions were based on clinical judgment. Various stents—plastic (straight (Amsterdam) or pig-tail, 7–10 Fr, 7–10 cm), covered or uncovered, and metallic (4–6 cm)—were used as indicated. Stents were scheduled for removal 30 days after ERCP or leak resolution (removed 30–60 days after leak resolution). Fluoroscopy was routinely used to verify the absence of retroperitoneal gas. No prophylactic pancreatic stents or NSAIDs were administered. All patients received IV fluids before and after ERCP, with an initial 1 L of IV fluids—either normal saline or Ringer’s lactate depending on patient status—administered peri-procedurally. All patients received prophylactic IV antibiotics (cefuroxime, ciprofloxacin, or piperacillin/tazobactam) tailored to allergies and previous exposures. Antiplatelet and anticoagulant therapy protocols evolved over time: prior to 2010, aspirin and clopidogrel were discontinued 5 days prior, and warfarin was managed with bridging therapy; post-2010, aspirin was generally continued, and DOACs were paused 48–72 h before ERCP, with INR monitored to ensure levels below 1.5. Sedation protocols included midazolam with or without fentanyl initially, with titrated propofol administration based on patient status.

### 2.4. Bile Duct Injury Classification

A completely transected CBD that led to the loss of biliary tree continuity and did not allow the upstream passage of the guidewire and visualization of the biliary tree was considered as a major leak. All others were considered minor leaks. Later on, bile leaks post-cholecystectomy were classified by the Strasberg classification for bile duct injury [17].

A right aberrant duct injury was documented in cases where no obvious leak was found in fluoroscopy during ERCP, which led to the re-appraisal of pre-ERCP and MRI-MRCP data or to the execution of a post-ERCP MRI-MRCP, which confirmed the existence of a right aberrant duct.

In Figure 2, a completely dissected common bile duct is presented. The cholangiographic image was obtained from a 37-year old female patient who presented with a bile leak post-laparoscopic cholecystectomy and was submitted to endoscopic retrograde cholangiopancreatography in our department.

### 2.5. Definition of Complications

Post-ERCP Pancreatitis: Serum amylase >3× normal with abdominal pain >24 h post-procedure [4].Perforation: Clinically significant contrast leakage, subcutaneous emphysema, or severe patient distress requiring intervention [5].

### 2.6. Bile Duct Healing

Bile duct healing was defined as the complete resolution/arrest of bile flow into drain reservoirs. In case of intrabdominal bile collections, bile duct healing was defined as a documented reduction in/resolution of the collection according to radiological studies.

### 2.7. Statistical Analysis

Data were analyzed using SPSS v27. Continuous variables were expressed as the mean ± SD or median (interquartile range). Comparisons utilized Student’s t-test or the Mann–Whitney U test; categorical variables used chi-squared or Fisher’s exact tests. Associations were assessed via Spearman’s correlations. Multivariate logistic regression models identified independent predictors. Only variables with *p* < 0.10 in the univariate analysis were entered into multivariate models. Goodness of fit was assessed by the Hosmer–Lemeshow test.

As this analysis involved de-identified retrospective data, ethics approval was not deemed necessary.

The authors read the STROBE Statement checklist of items, and the manuscript was prepared and revised according to this checklist.

## 3. Results

### 3.1. Patient Demographics and Bile Leak Characteristics

A total of 176 patients (91 males and 85 females; mean age 62 ± 16 years) underwent ERCP for bile leaks. Post-cholecystectomy was the most common etiology of bile leak injury, with the majority of them being laparoscopic procedures, followed by hepatectomy and surgery for hydatic cyst disease. In nine cholecystectomies, CBD exploration was also undertaken during the operation (eight during open cholecystectomy and one during laparoscopic cholecystectomy) (Table 1). A documented intra-abdominal bile collection was present in nine patients (5.1%), while the median daily output of bile in patients with drainage was 300 cc (min 50–max 1500 cc). The mean interval from bile leak appearance to ERCP was 7 days (min 1–max 72 days).

### 3.2. Cannulation Success and Cholangiographic Findings

CBD cannulation was achieved during the first ERCP in 97.2% of patients (171/176). Among patients with an intact papilla (without previous sphincterotomy), successful cannulation occurred in 96.9% (157/162), with 63/162 (38.9%) precut fistulotomies during initial cannulation.

Cholangiography revealed complete CBD division in 10 (5.7%) patients, who were considered to have major leaks and referred for surgery. The remaining 166 patients had minor leaks. In four patients, the exact leak point could not be identified. In patients where the leak site was documented (n = 162), in 65/162 (40.1%), it originated from the cystic duct stump, in 39/162 (24.1%) from an injured but not completely clipped CBD, in 25/162 (15.4%) from the gallbladder bed, in 23/162 (14.2%) from branches of the right or left biliary tree, in 9/162 (5.6%) from an aberrant right duct, and in 1/162 from the cholecystostomy site.

Isolating patients with a cholecystectomy-related bile leak injury (n = 140/143; in 3/143 patients, the identification of the exact bile leak site was not possible), the Strasberg classification was as follows: Strasberg A in 87/140 (62.15), C in 7/140 (5.0%), D in 38/140 (27.1%), and E in 8/140 (5.7%).

A stricture was identified in 29 patients (16.5%), comprising primarily low- or mid-CBD strictures (n = 13, 59.1%), mostly attributed to a clip injury during cholecystectomy. Strictures at the biliary bifurcation were observed in seven patients (31.8%), and isolated hepatic duct stenosis appeared in two patients (9.1%).

Stones were detected in 21 patients (11.9%), mostly under 1.5 cm in diameter (<1.5 cm in 19/21, 90%). Complete CBD clearance from stones was achieved in 18 of these (85.7%), with partial clearance in two (9.5%) and failure in one patient (4.8%).

### 3.3. Healing Rates

Among patients with minor leaks (n = 166, with post-ERCP follow-up data for 160), leak resolution was achieved in 145 patients (90.6%).

Patients’ age did not differ significantly between those with leak resolution and those without (62 ± 17 years vs. 61 ± 18 years; *p* = 0.845). Similarly, there was no significant difference in healing rates based on sex (*p* = 0.479). The presence of retained stones did not influence the likelihood of leak closure (*p* = 0.659), nor did the existence of a stricture on cholangiography (*p* = 0.495).

The highest rates of leak resolution were observed among patients with leaks originating from the cystic duct stump, whereas the lowest rates were seen in patients with leaks from branches of the right or left biliary tree. Additionally, bile leaks following cholecystectomy showed a trend toward higher healing rates compared to leaks from other hepatopancreatobiliary procedures, although this difference did not reach statistical significance (*p* = 0.109) (Table 2).

Specifically, in patients with post-cholecystectomy bile leaks, patients with a Strasberg A bile duct injury showed numerically higher healing rates, but not in a statistically significant manner, when compared to patients with Strasberg C or D injuries (83/87, 95.4% vs. 38/43, 88.4%, *p* = 0.133)

Healing rates were also not significantly affected by the presence of intra-abdominal fluid collections: 7/8 (87.5%) in patients with collections versus 123/130 (92.3%) in those without (*p* = 0.487). Furthermore, the pre-ERCP daily volume of bile drainage did not differ between patients with healed leaks and those with ongoing leaks (402 ± 277 cc/day vs. 333 ± 152 cc/day; *p* = 0.674).

### 3.4. Stent Application

Stents were placed in 135/166 patients with a minor leak (bile leak healing data available in 130/135). Plastic stents were applied in 123/166 (74.1%) patients with a minor leak. Straight (Amsterdam-type) stents were used in 101/123 (81.1%) cases with plastic stent placement, and pig-tail stents were used in 22/123 (17.9%). Nasobiliary drainage catheters were used in 5/166 and metal stents (covered or uncovered) in seven (4.2%).

Stent placement was associated with a higher healing rate (*p* = 0.007); see Table 2. The success rate of pig-tail stents (18/20, 90%) did not differ significantly from that of straight stents (95/99, 96%; *p* = 0.267). Healing rates among patients with metal stents and nasobiliary catheters were 85.7% and 80%, respectively.

In the multivariate analysis, stent application was independently associated with leak resolution (OR 4.515; 95% CI 1.432–14.239; *p* = 0.010), whereas the site of the leak and the HPB operation that lead to the leak were not (OR 0.892; 95% CI 0.608–1.050; *p* = 0.108 and OR 0.715; 95% CI 0.684–1.158; *p* = 0.385).

For the 15 patients with unsuccessful leak closure, three (20%) died (one during a second ERCP due to cardiac arrest and two from sepsis), while the remaining 12 underwent alternative treatments such as percutaneous transhepatic cholangiography (PTC) or surgery.

### 3.5. ERCP Needed Until Healing

In 11/166 patients with a minor leak (6.6%), a subsequent ERCP was performed due to initial failure. In nine patients, only one subsequent ERCP was performed. In one patient, in total, four ERCPs were executed (unsuccessful initial cannulation, CBD stricture existence, repetitive dilations, and, finally, metal stent placement); meanwhile, in another patient, a total of three ERCPs were needed as the leak healed only after using a nasobiliary drainage catheter (injured right aberrant). Regarding the other nine patients in whom two ERCPs were needed in order for the leak to heal, in 3/9, a stent was initially not placed and the leak was healed only after stent placement in the subsequent ERCP. In 5/9, in the initial ERCP, CBD cannulation was not effective but succeeded during the second attempt, while, in 1/9, a middle stricture due to CBD partial clipping was observed, and a second ERCP was needed in order to dilate the stricture; a stent was also needed as the initial attempt failed.

### 3.6. Time to Leak Healing

The median time from the initial ERCP to leak resolution was 2 days (range: 1 to 35 days). Patients who underwent cholecystectomy tended to require less time to achieve leak healing after ERCP compared to those with leaks following other HPB surgeries: a median of 2 days (min 1–max 35 days) versus 3 days (min: 1–max 19 days), *p* = 0.065. Moreover, when comparing patients submitted to cholecystectomy, patients with a Strasberg A bile duct injury showed faster leak resolution when compared to patients with a Strasberg C but not D bile duct injury (2 days (min 1–max 15 days) versus 10 days (min 2–max 35 days) and 2 days (min 1–max 25 days), *p* < 0.001 and *p* = 0.207, respectively).

The longest healing times were observed in patients with leaks originating from a right aberrant duct. Specifically, patients with right aberrant bile leaks experienced longer time to healing compared to those with leaks from the cystic duct stump, a CBD injury, or leaks from the gallbladder bed (Figure 3). The time to healing did not differ between other commonly observed bile leak etiologies.

The time to leak resolution after ERCP did not significantly differ in patients with cholangiographic strictures, even when numerically longer (3 days (min 1–max 25 days) vs. 2 days (min 1–max 35); *p* = 0.202), or among patients with concomitant stones (2 days (min 1–max 10 days) vs. 2 days (min 1–max 35 days); *p* = 0.897).

Similarly, the time to leak healing was not significantly different between patients in whom a stent was placed and those managed without stenting (2 days (min 1–max 35 days) vs. 2 days (min 1–max 19 days); *p* = 0.842). No correlation was found between the daily output of bile in the drainage and the time needed for leak resolution (correlation coefficient: r = 0.133; *p* = 0.348).

### 3.7. Timing of ERCP and Leak Resolution

Patients whose leaks healed were referred for ERCP earlier, with a median time of 7 days (min 1–max 72) versus 9 days (min 3 days–max 42 days) among those with incomplete healing; however, this difference was not statistically significant (*p* = 0.285). Importantly, the later the ERCP was performed, the longer it took for the leak to resolve (r = 0.362; *p* < 0.001).

When dividing the cohort into two groups—patients treated within the first week after surgery or trauma and those treated later—patients in whom ERCP was postponed beyond one week showed a trend toward lower healing rates (88.3% vs. 95.5%), but this difference was not statistically significant (*p* = 0.121). Nonetheless, patients in whom ERCP was postponed beyond one week were characterized by significantly longer times to leak closure (3 days (min 1–max 25 days) vs. 1 day (min 1–max 15 days); *p* < 0.001).

When once again dividing the cohort into two groups—patients treated within the first 3 days after surgery or trauma and those treated later—patients in whom ERCP was postponed beyond 3 days showed a trend toward lower healing rates (93% vs. 100%), but this difference was not statistically significant (*p* = 0.360). The time to bile leak resolution did not differ between patients in whom ERCP was postponed beyond 3 days and those in whom it was not (2 days (min 1–max 25 days) vs. 2 day (min 1–max 15 days); *p* = 0.888).

When dividing the cohort into two periods—namely, period A for patients treated until 12/12 (n = 83) and period B for those after 1/13 (n = 77)—no differences in healing rates were found (period A: 74/83 (89.2%) vs. period B: 71/77 (92.2%), *p* = 0.350). No differences concerning stent placement were noted (period A: 70/83 (84.3%) vs. period B: 65/81 (80.2%), *p* = 0.315). Additionally no differences were noted in the median time to ERCP from bile leak diagnosis (period A: 8 days (min 1–max 42 days) vs. period B: 7 days (min 1–max 72 days); *p* = 0.653) or resolution of the bile leak from the time of ERCP (period A: 2 days (min 1–max 25 days) vs. period B: 2 days (min 1–max 19 days); *p* = 0.758).

### 3.8. Complications

ERCP-related adverse events were rare overall: post-ERCP pancreatitis occurred in two patients (1.14%) and retroperitoneal perforation occurred in one (0.5%) (endoscopic treatment with placement of a fully covered metallic stent was applied), and there was one ERCP-related death (0.5%) due to sedation-related cardiac arrest.

## 4. Discussion

In this large, real-world series of 176 consecutive patients treated with ERCP for bile leak, our findings reinforce the notion that transpapillary stent placement should be the initial endoscopic strategy for minor bile leaks. Overall, ERCP achieved leak resolution in 90.6% of patients with a minor leak, and stent placement was associated with higher closure rates than endoscopic management without stent placement. Our study’s healing rate is in agreement with previous results showing healing rates close to 90% [16]. In the multivariate analysis, stent application remained an independent predictor of leak resolution among patients with a minor bile leak, concordant with guideline recommendations and prior series that emphasize stenting to both decompress the biliary system and promote the sealing of focal disruptions [6,17,18]. Additionally, we observed no significant difference in healing rates between straight (Amsterdam) and pig-tail plastic stents, although our results were derived from a small number of cases in whom a pig-tail stent was inserted—a fact that may affect the generalization of our findings. This aligns with prior reports suggesting that the stent diameter or configuration exerts a limited influence on short-term leak resolution provided that effective transpapillary drainage is achieved [19,20].

Our data indicate that earlier referral for ERCP does not significantly change the overall probability of leak healing, but, according to our analysis, it may be associated with a shorter interval to clinical resolution among patients with a minor bile leak. Although the absolute healing rates were similar when ERCP was performed within versus after the first postoperative week (95.5% vs. 88.3%; *p* = 0.121), the time to leak closure was significantly shorter with earlier ERCP. In our study, the effect of ERCP timing in the acceleration of bile leak clinical resolution was only observed when comparing patients submitted to ERCP during the first week after bile leak diagnosis or later, and it was not obvious among patients submitted to ERCP within the first 3 days from diagnosis or not. Once again, in our study, as also found in the studies by Desai et al. and Chen et al., it was shown that submitting a patient with a bile leak to ERCP in the first three days from diagnosis does not change either the healing rate or the interval to resolution [21,22]. Nonetheless, in our study, we did find a difference concerning the time to leak closure if ERCP was executed within the first week from bile leak diagnosis, as well as a numerical trend showing higher healing rates. This trend was also shown in the study by Chen et al. (88.9% vs. 82.9%). Chen et al., however, did not assess the time to bile leak resolution, so no comparison between the two studies can be made. Still, it should be underlined that the authors found a significant difference concerning healing rates when comparing patients submitted to ERCP within the first 3 weeks from bile leak diagnosis or later. Thus, while early ERCP may not increase the ultimate success rates among patients with a minor bile leak, prompt endoscopic decompression, according to our data, may accelerate recovery and shorten the period of external bile drainage—a clinically relevant outcome regarding patient comfort, hospital stays, and resource utilization [21,22,23].

The leak origin affected both the speed and likelihood of resolution among patients with a minor bile leak. Cystic duct stump leaks demonstrated the highest healing rates. Specifically, for patients with a post-cholecystectomy bile leak, patients with a Strasberg A injury showed increased rates and faster resolution of the bile leak after ERCP, an observation that is in agreement with previous studies showing excellent results for the endoscopic treatment of patients with Strasberg A bile duct injuries [24,25]. Additionally, according to our data, leaks from aberrant right hepatic branches healed more slowly despite similar eventual closure rates to other minor leaks. This observation suggests that, in such cases, patience is essential, as additional intervention is not routinely necessary, unless clinical deterioration occurs. In our opinion, based on our findings—despite arising from only nine patients—it is likely that ERCP does not provide any significant benefit in patients with a leak originating from a right aberrant duct, as any changes in biliary tree pressure resulting from the intervention will not affect leaks originating from the segment of the liver that is drained by the aberrant duct. This underlines the value of pre-ERCP imaging with MRCP in patients with a bile leak, as an intervention such as ERCP will likely expose the patient to an increased risk without significant benefits.

The presence of retained stones or cholangiographic strictures did not materially alter the time to healing or healing rates in our cohort, suggesting that successful endoscopic decompression and sealing via stenting can overcome the additional upstream pressure effects imposed by stones or focal strictures once they are addressed endoscopically. Similarly, the pre-EPCP volume of external biliary drainage did not predict either the likelihood of closure or the interval to healing, indicating that the absolute drainage output alone should not be used to triage patients away from or toward endoscopic therapy.

Endoscopic management in this series carried a low complication burden: post-ERCP pancreatitis was uncommon (1.1%), retroperitoneal perforation occurred in 1.1%, and ERCP-related mortality was 0.5%. These event rates compare favorably with contemporary series [26,27]. Additionally, they support the role of ERCP as a safe first-line intervention for minor bile leaks when performed in experienced centers [1,28]. The low overall adverse event rate observed here further argues in favor of an endoscopy-first algorithm in appropriate patients.

Several limitations warrant emphasis. Although data were prospectively collected, the analysis was retrospective and subject to inherent biases. The 25-year study interval encompassed significant evolution in ERCP devices, sedation practices, and peri-procedural care, which may have influenced both technical success and complication rates. However, concerning the sedation protocol, our department’s protocol has not changed over the years. Regarding stenting, the first wall stent (metallic stent) was placed in our department on September 2007; therefore, since the first ERCP due to bile leak included in our study occurred on January 2002, in only a minority of cases, metallic stents were not available (in total, 35 cases). Thus, no comparisons can be made as, during the longest period of the study, metallic stents were available. Procedural decisions, including stent selection and timing, were guided by operator judgment rather than a prespecified protocol, introducing potential selection bias. Unfortunately, no data about patients’ clinical severity, as well as critical predictors of bile duct injury—surgeon experience, operative difficulty, and acute inflammation presence—are available in our database and therefore no analysis/propensity matching could be performed. The study was performed at two tertiary HPB centers under the care of a single senior endoscopist and trainees, which may limit its generalizability to lower-volume settings. Finally, although we report measures of clinical resolution and the time to healing, we lacked granular data on patient-reported outcomes and the cost—both for patients and for the healthcare system—outcomes that are important when assessing the broader impacts of earlier intervention. Moreover, since our hospital is a referral center and patients were hospitalized in other institutions, attending our hospital only for the procedure and then returning to their main institution, post-ERCP follow-up data were missing in some cases, especially on long-term clinical outcomes. Additionally, no surgical comparison group was available in our cohort and therefore no comparisons concerning efficacy, complications, and cost between ERCP and surgery could be performed.

Our findings support the current guideline recommendations that favor transpapillary stenting for minor postoperative bile leaks and provide new real-world evidence that earlier ERCP may shorten the time to clinical resolution, even when the overall healing rates are unchanged [1,28]. Clinicians should therefore consider prompt endoscopic evaluation and stent placement for patients with persistent postoperative biliary drainage or imaging-documented leaks, reserving surgical repair or percutaneous approaches for major duct transections or endoscopic failures. For leaks originating from aberrant right hepatic branches, our data suggest that clinicians can reasonably expect eventual closure but should counsel patients regarding a longer expected time to resolution and maintain vigilance for signs that warrant escalation of care.

Prospective studies are needed to more definitively quantify the tradeoffs between immediate and delayed ERCP with respect to the hospital length of stay, costs, quality of life, and complications. Pragmatic trials or prospective registries that capture standardized anatomic leak classification, the timing of intervention, device selection, and patient-centered outcomes (length of stay, need for additional procedures, pain scores, and health-economic measures) are needed to define optimal, cost-effective care pathways for different leak types.

## 5. Conclusions

In this large, real-world cohort, ERCP with transpapillary stent placement achieved high overall healing rates for minor bile leaks and was an independent predictor of successful closure. The execution of ERCP within the first week following the surgery that led to the bile leak was not associated with higher ultimate healing rates but significantly shortened the time to clinical resolution. The leak origin influenced the speed of healing (with cystic duct stump leaks resolving most rapidly and aberrant right hepatic duct leaks taking longer), whereas the presence of retained stones, cholangiographic strictures, or higher external drainage volumes did not materially affect outcomes once effective endoscopic management was achieved. These findings support an endoscopy-first strategy—with the early consideration of transpapillary stenting—for most patients with minor postoperative bile leaks, while reserving surgical or percutaneous approaches for major duct transections or ERCP failures.

## Figures and Tables

**Figure 1 medicina-61-02108-f001:**
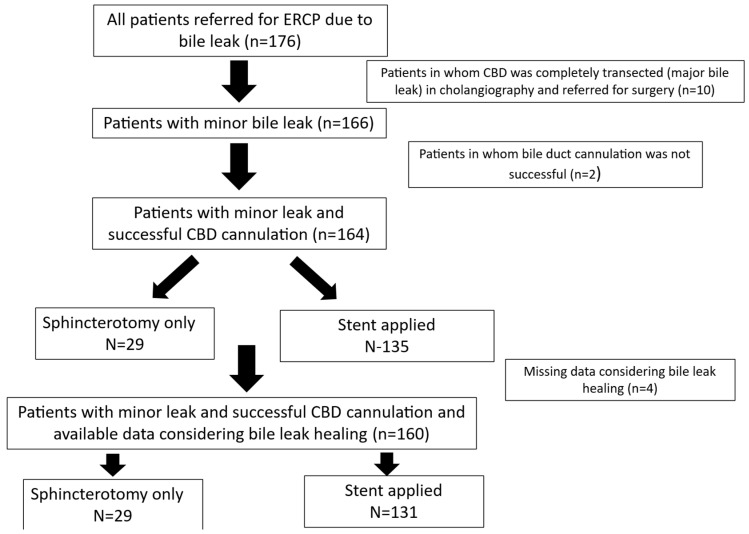
Study flowchart.

**Figure 2 medicina-61-02108-f002:**
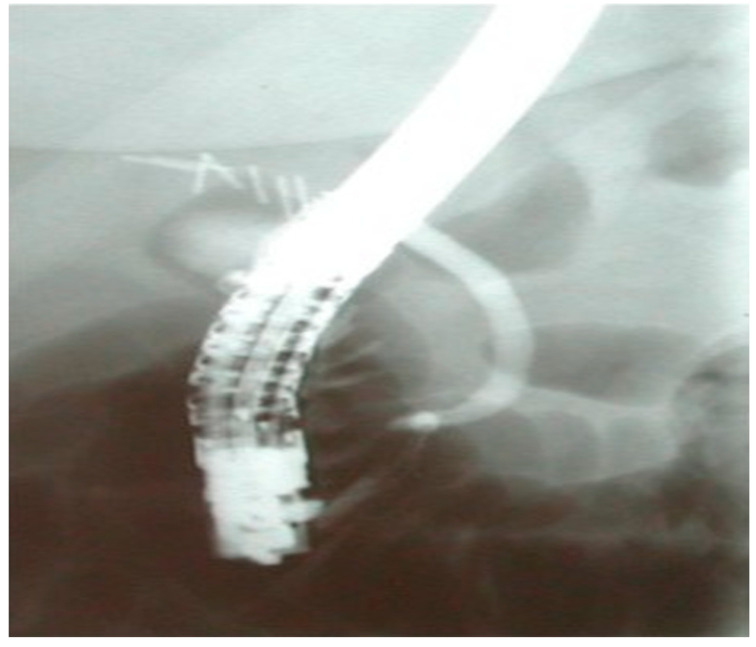
A completely dissected CBD post-cholecystectomy.

**Figure 3 medicina-61-02108-f003:**
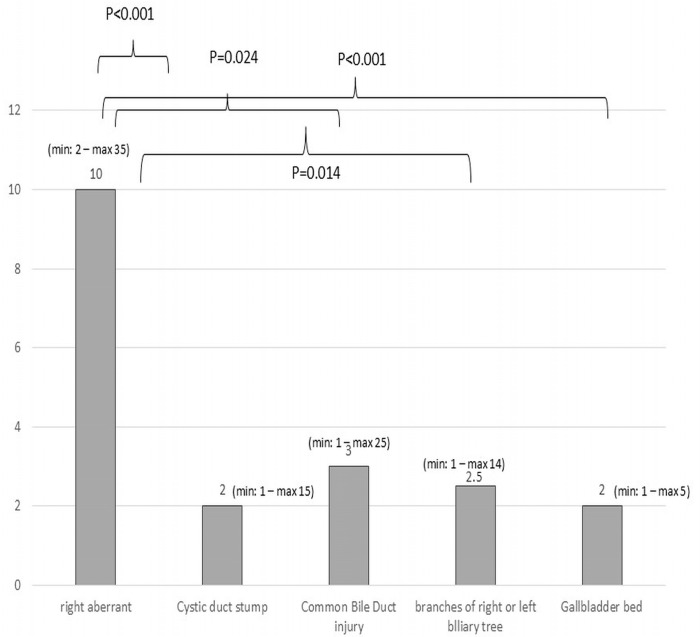
Healing times for different bile leak sites.

**Table 1 medicina-61-02108-t001:** Type of intervention/event that led to bile leak.

Intervention/Event	N (%)
Cholecystectomy Laparoscopic Open Converted to open from lap	143/176 (81.3%) 87/176 (49.4%) 53/176 (30.1%) 3/176 (1.7%)
Hepatectomy	18/176 (10.2%)
Liver surgery for hydatid cyst disease	8/176 (4.5%)
Liver trauma	2/176 (1.1%)
Radio-frequency ablation of liver metastasis	1/176 (0.6%)
Whipple’s procedure	1/176 (0.6%)
Bilioenteric bypass	1/176 (0.6%)
Liver transplantation	1/176 (0.6%)
Cholecystostomy	1/176 (0.6%)

**Table 2 medicina-61-02108-t002:** Bile leak healing rates according to patient characteristics, cholangiographic findings, and stent application.

	Bile Leak Healing Rate N, %	*p*
Sex		0.479
Male	71/79, 90%	
Female	74/81, 90.5%	
Operation		0.109
Cholecystectomy	121/131 (92.4%)	
Other	24/29 (82.8%)	
Bile leak site of origin		**<0.001**
Cystic duct stump	63/65, 96.9%	
CBD injury	34/39, 87.2%	
Gallbladder bed	22/25, 88.0%	
Branches of left or right biliary tree	18/21, 85.7%	
Right aberrant	8/9, 88.9%	
Cholecystostomy site	0/1, 0%	
Stricture existence in cholangiography		0.603
Yes	22/24, 91.7%	
No	123/136, 90.4%	
Stone existence		0.616
Yes	19/21, 90.5%	
No	126/139, 90.6%	
Stent application		**0.007**
Yes	123/131, 93.9%	
No	22/29, 75.9%	

*p* statistically significant if <0.05, CBD: common bile duct. Bold: It defines statistical significance compared to the other variables.

## Data Availability

The original contributions presented in this study are included in the article. Further inquiries can be directed to the corresponding author.

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
