# Peer review of "Bile Leak: Is There Optimal Timing for Endoscopy?"

_medicina, 2025, doi:10.3390/medicina61122108_

Round 1

Reviewer 1 Report (Previous Reviewer 2)

Comments and Suggestions for Authors

All the issues have been addressed 

The manuscript has been modified and reads better now

Author Response

Dear Reviewer,

We would like to sincerely thank you for your insightful comments, which have helped improve the quality and clarity of our manuscript.

Reviewer 2 Report (Previous Reviewer 3)

Comments and Suggestions for Authors

How well does ERCP handle bile leakage after hepatopancreatobiliary surgery?

When treating bile leaks with varying degrees of biliary duct disruption, how does ERCP compare to surgical repair?

How accurate are ERCP and MRCP in diagnosing and treating bile leak sites?

What effect do stent placement and timing have on the healing rates of patients having endoscopic retrograde cholangiopancreatography (ERCP) for bile leaks of various etiologies?

In addressing bile leaks across different clinical presentations, how do healing rates vary between early (within 3 days) and delayed ERCP, as well as between stent implantation and sphincterotomy alone?

In patients with various underlying biliary duct injury features, what are the specific healing rates and problems related to transpapillary stenting for bile leaks?

Please provide the number of attempts that met the criteria for "multiple unsuccessful attempts" prior to the needle-knife precut.

Describe whether the choice between metallic and plastic stents was determined by operator preference alone or by clinical standards.

If available, describe the kind and frequency of peri-procedural antibiotic adverse effects.

Author Response

Dear Reviewer,

Thank you very much for taking the time to review our manuscript and for providing valuable comments and suggestions. Below you may find our detailed responses to the comments. Your comments are presented in regular font, and our responses follow in italics.

Comment 1

How well does ERCP handle bile leakage after hepatopancreatobiliary surgery?

Response

With respect to your valuable comment but that is exactly what our article is about. Specific data are given in our result part while as vividly stated in our conclusion ‘’ Conclusion In this large, real‑world cohort, ERCP with transpapillary stent placement achieved high overall healing rates for minor bile leaks and was an independent predictor of successful closure. Execution of ERCP during the first week from the surgery that led to bile leak was not associated with higher ultimate healing rates but significantly shortened the time to clinical resolution. Leak origin influenced the tempo of healing (with cystic duct stump leaks resolving most rapidly and aberrant right hepatic duct leaks taking longer), whereas the presence of retained stones, cholangiographic strictures, or higher external drainage volume did not materially affect outcomes once effective endoscopic management was achieved’’ Moreover a more detailed discussion about ERCP efficacy in handling bile leakage after HPB surgery is provided in our discussion part.

Comment 2

When treating bile leaks with varying degrees of biliary duct disruption, how does ERCP compare to surgical repair?

Response

Thank you for this insightful question. Unfortunately, a cohort of patients in whom surgical correction was applied is not available in our study as it origins from an endoscopy department. Though as stated in our result part and once again underlined in our discussion part (line 4); Overall ERCP achieved leak resolution in 90.6% of minor‑leak cases.

Comment 3
How accurate are ERCP and MRCP in diagnosing and treating bile leak sites?

Response

Thank you for pointing this out. Our study does not address this question which is outside the scope of our manuscript. MRCP according also to guidelines on the topic (ref 6 of our manuscript) is the gold standard for bile leak diagnosis but cannot be used a therapeutic modality while ERCP can.

Comment 4

What effect do stent placement and timing have on the healing rates of patients having endoscopic retrograde cholangiopancreatography (ERCP) for bile leaks of various etiologies?

Response

With respect to your insightful comment but that is exactly what our article is about. Specific data are given in our result part while as vividly stated in our conclusion ‘’ Conclusion In this large, real‑world cohort, ERCP with transpapillary stent placement achieved high overall healing rates for minor bile leaks and was an independent predictor of successful closure. Execution of ERCP during the first week from the surgery that led to bile leak was not associated with higher ultimate healing rates but significantly shortened the time to clinical resolution. Leak origin influenced the tempo of healing (with cystic duct stump leaks resolving most rapidly and aberrant right hepatic duct leaks taking longer), whereas the presence of retained stones, cholangiographic strictures, or higher external drainage volume did not materially affect outcomes once effective endoscopic management was achieved’’.

Comment 5

In addressing bile leaks across different clinical presentations, how do healing rates vary between early (within 3 days) and delayed ERCP, as well as between stent implantation and sphincterotomy alone?
Response

Thank you for pointing this out. In section 3.7, lines 296-301 data about healing rates between early and delayed ERCP can be found. Data about healing rates in patients with stent or only sphincterotomy can be found on table 2

Comment 6
In patients with various underlying biliary duct injury features, what are the specific healing rates and problems related to transpapillary stenting for bile leaks?

Repsonse

Thank you for pointing this out. Healing rated for specific bile duct injuries are provided in table 2.

Comment 7

Please provide the number of attempts that met the criteria for "multiple unsuccessful attempts" prior to the needle-knife precut.

Response

In response to your suggestion, we have now clarified, that multiple unsuccessful attempts was defined as >5  (paragraph 2.3 , line 110).

Comment 8

Describe whether the choice between metallic and plastic stents was determined by operator preference alone or by clinical standards.

Response

Thank you for this valuable comment. Stent type was determined by operator clinical judgement as stated already in our method section. As stated in our results metallic stent were applied  in only 7 patients (1 due to perforation, 1 due to previous failure of a plastic stent application, both cases described in our result section and in other 5 patients with strictures) .

Comment 9

If available, describe the kind and frequency of peri-procedural antibiotic adverse effects.

Response

All patients as presented with bile leak post an HPB surgery were already on antibiotics. Unfortunately data about antibiotic adverse effect were not available.

Kind Regards,

This manuscript is a resubmission of an earlier submission. The following is a list of the peer review reports and author responses from that submission.

Round 1

Reviewer 1 Report

Comments and Suggestions for Authors

This retrospective study examines ERCP timing effects on bile leak outcomes through a substantial 25-year cohort. While addressing a clinically relevant question, the work contains fundamental flaws that preclude publication.

Major Comments

1. The 25-year period encompasses major changes in surgical techniques, ERCP technology, and patient selection without analytical adjustment. Patients receiving early versus delayed ERCP likely differ systematically in clinical severity and era of treatment. No propensity matching or time-stratified analysis addresses these confounders.
2. "Early" ERCP varies between 3 days and 1 week without justification. The primary endpoint "leak healing" lacks standardized criteria - cessation of drainage, radiographic resolution, or clinical improvement remains undefined. This precludes meaningful interpretation and replication.
3. Multivariate analysis lacks essential details including variable selection, model building, and assumption testing. The correlation coefficient (r=0.362) indicates only moderate association yet is presented as definitive. No effect sizes accompany p-values, and missing data patterns are not analyzed.
4. Critical predictors of bile duct injury - surgeon experience, operative difficulty, acute inflammation presence, conversion rates - are not systematically analyzed. This represents the study's greatest limitation as these factors actually determine outcomes.

Minor Comments

1. ERCP procedures (lines 103-122) lack sufficient detail for replication. Stent removal criteria are vague.
2. Discussion inadequately positions findings within recent systematic reviews and current guidelines.
3. The claim that "ethics approval was not required" (lines 151-152) may not meet current standards for retrospective studies.
4. Healthcare costs are mentioned but no data provided - either include analysis or explicitly acknowledge this limitation.

Author Response

Comment 1
The 25-year period encompasses major changes in surgical techniques, ERCP technology, and
patient selection without analytical adjustment. Patients receiving early versus delayed ERCP
likely differ systematically in clinical severity and era of treatment. No propensity matching or
time-stratified analysis addresses these confounders.
Response
Concerning the effect of time on ercp technology, this limitation is already acknowledged in
our study limitation part, which is now enriched.
Moreover in order to overcome reviewers concerns regarding possible selection bias arising
from the large period of our study we added a small paragraph in our result part comparing
healing rates as also stent usage, time to ERCP and time to bile leak resolution, after dividing
patients related to the time of ERCP execution (before or after the year 2012) – last
paragraph section 3.7.
Once again, it was added in our study limitation part that; Unfortunately no data about
patients clinical severity are available in our database and therefore no analysis/propensity
matching can be performed (the amount of bile leak and the existence of an intrabdominal
collection – factors already analysed in our study cannot serve as clinical severity markers)
Comment 2.
"Early" ERCP varies between 3 days and 1 week without justification. The primary endpoint
"leak healing" lacks standardized criteria - cessation of drainage, radiographic resolution, or
clinical improvement remains undefined. This precludes meaningful interpretation and
replication.
Response
No definition of early ERCP is given in our manuscript. Indeed in the section of the result part
we divided patient either submitted to ERCP the first week or later but nowhere in our original
manuscript we did not used a cut-off of 3 days. A very recent study by Chen though assessed
both time limits (ref 22 in our manuscript and found no differences).We used 1 week time limit
as it was a limit previously used by Chen et all in their study and because according to our
clinical practice we use a 1 week criterion in order to proceed to ERCP if there is still a flow >
100ml in the surgical drain one week after an HPB surgery as already stated in the method
section .
In our revised manuscript though as another reviewer asked for the analysis concerning the
time limit of 3 days to be included, such an analysis has been now added. Still we did not
proceed to any definition concerning the time of ERCP and in the results as also discussion part
of our manuscript we describe and commentate our findings without proceed to further
definitions
Concerning bile duct healing it is now Defined now in method section 2.6 : ‘’ Bile duct healing
was defined as complete resolution/arrest of bile flow into drain reservoirs. In case of
intrabdominal bile collections, bile duct healing was defined as documented
reduction/resolution of the collection by radiological studies.
Comment 3.
Multivariate analysis lacks essential details including variable selection, model building, and
assumption testing. The correlation coefficient (r=0.362) indicates only moderate association
yet is presented as definitive. No effect sizes accompany p-values, and missing data patterns
are not analyzed.
Response
As stated now in our method part at the end of 2.7 paragraph: Multivariate logistic
regression models identified independent predictors. Only variables with P < 0.10 in
univariate analysis were entered into multivariate models. Goodness of fit was assessed by
the Hosmer-Lemeshow test. Concerning the correlation coefficient of r=0.362, indeed it
indicated only moderate association though our statement is further supported by the fact
that when dividing the cohort into two groups—patients treated within the first week after
surgery or trauma and those treated later— delayed ERCP was associated with a significantly
longer time to leak closure (3 days (min: 1 – max: 25 days) vs. 1 day (min: 1 – max 15 days);
p<0.001). Regarding missing data as stated now in our method section 2.2); ‘’Missing data
were handled through pairwise deletion, utilizing all available data points for each analysis
without imputation.’’
Comment 4.
Critical predictors of bile duct injury - surgeon experience, operative difficulty, acute
inflammation presence, conversion rates - are not systematically analyzed. This represents
the study's greatest limitation as these factors actually determine outcomes.
Response
Unfortunately no such data are available in our database and therefore no
analysis/propensity matching can be performed. A comment was added in our study
limitation part
Minor Comments
1 ERCP procedures (lines 103-122) lack sufficient detail for replication. Stent removal
criteria are vague.
Response
As stated in our method section ‘’ Stents were scheduled for removal 30 days after ERCP or
leak resolution (removed 30–60 days after leak resolution’’ With respect to the reviewers
opinion this is something specific and not vague. Additionally ERCP procedure is clearly and in
detail described in the method section
2. Discussion inadequately positions findings within recent systematic reviews and
current guidelines.
Response
Discussion is now enriched. Unfortunately no systematic reviews and meta-analysis exist in
the field when focusing on ERCP efficacy and what the best timing to ERCP from bile leak
diagnosis exist but only small in most cases retrospective studies and simple reviews . Still
concerning timing to ERCP, our findings are now more analytically commentated in
conjunction to findings from previous studies.
3. The claim that "ethics approval was not required" (lines 151-152) may not meet
current standards for retrospective studies.
Response
As stated in our method section ‘’ As this analysis involved de-identified retrospective data,
ethics approval was not deemed necessary.’’
4. Healthcare costs are mentioned but no data provided - either include analysis or
explicitly acknowledge this limitation.
Response
It is already acknowledged as one of our articles limitation, though new added a small phrase
in order to highlight it.

Reviewer 2 Report

Comments and Suggestions for Authors

A retrospective study, with its inherent limitations, nevertheless, adds to the data on this subject

The data has been curated well

It has been written adequately

The review of literature is comprehensive

The following are suggested for the manuscript's betterment:

  1. STROBE guidelines to be followed. The same should be mentioned in the methods section
  2. Please highlight the take-home message of the study

Author Response

Comment 1
STROBE guidelines to be followed. The same should be mentioned in the methods section
Response
It was added in the method section that: The authors have read the STROBE Statement-
checklist of items, and the manuscript was prepared and revised according to the STROBE
Statement-checklist of items. A strobe statement is now included in our study documents
(pages where each part of the statement can be found are highlighted in yellow in the
statement)
Comment 2
Please highlight the take-home message of the study
Response
The take home message of our study is presented in the last sentence of the conclusion of our
manuscript in which is vividly stated that ‘’ These findings support an endoscopy first
strategy—with early consideration of transpapillary stenting—for most patients with minor
postoperative bile leaks, while reserving surgical or percutaneous approaches for major duct
transections or ERCP failures.’’ More analytically it is stated in

Reviewer 3 Report

Comments and Suggestions for Authors

To enhance the manuscript's quality, the author recommended addressing the following comment.

Which preventative and diagnostic measures are most successful in lowering the risk of bile leak problems following hepatopancreatobiliary surgery?

What effects do various surgical methods have on the frequency and intensity of bile leaks in patients who have had cholecystectomy?

How do postoperative bile leaks in patients receiving hepatopancreatobiliary surgeriesaffect their long-term clinical and financial outcomes?

In comparison to surgical correction, how effective is ERCP as a treatment for bile duct leaks?

In detecting and treating bile duct injuries, how does ERCP's diagnostic and therapeutic methodology stack up against MRCP?

What effect does the lowering of transpapillary pressure gradient caused by ERCP have on the healing of bile ducts in individuals who have biliary tract injuries?

When it comes to treating bile leaks, when and how should ERCP be performed?

What effects do various ERCP procedures, such as sphincterotomy and stent implantation, have on the rate at which bile leak patients heal?

When compared to delayed intervention, does early ERCP (within three days) improve healing outcomes for individuals with bile leaks?

When treating surgical bile leakage in patients referred from two busy hepato-pancreato-biliary clinics in Greece, how accurate is ERCP in terms of diagnosis and clinical efficacy?

What is the relationship between the identification and treatment of postoperative bile leakage and the various ERCP referral criteria (drain output thresholds and imaging findings)?

In treating postoperative bile leaks, what are the procedural results and complication rates of ERCP carried out by senior endoscopists as opposed to supervised trainees?

What are the safety profiles and procedural results of various stenting and cannulation approaches in ERCP?

What effects do various sedation and antiplatelet management techniques have on the efficacy and safety of ERCP procedures?

What modifications were made to stenting and anticoagulation techniques in ERCP operations between the pre-2010 and post-2010 periods?

Author Response

To enhance the manuscript's quality, the author recommended addressing the following
comment.
Comment
Which preventative and diagnostic measures are most successful in lowering the risk of bile
leak problems following hepatopancreatobiliary surgery?
Answer
Those questions cannot unfortunately be answered by our study which is a real world study
focusing on the effectiveness of ercp in treating bile leak.
Comment
What effects do various surgical methods have on the frequency and intensity of bile leaks in
patients who have had cholecystectomy?
Answer
Our study is an endoscopic study focusing on ERCP and its place in the management of bile
leak and not a surgical study focusing on surgical methods correlated to the frequency and
the intensity of the leak. Since our data origin from an endoscopy department we are unable
to answer to such question with the current study
Comment
How do postoperative bile leaks in patients receiving hepatopancreatobiliary surgeries affect
their long-term clinical and financial outcomes?
Answer
Long term data are not available on our study which focus on bile leak resolution. A
statement was added in the study limitation group mentioning this exact limitation (last
phrase; ‘’ especially long term clinical outcomes. ‘’ In the study limitation part though is
already stated that; Finally, although we report measures of clinical resolution and time to
healing, we lacked granular data on patient reported outcomes, and cost, outcomes that are
im-portant when assessing the broader impact of earlier intervention
Comment
In comparison to surgical correction, how effective is ERCP as a treatment for bile duct leaks?
Response
Unfortunately, a cohort of patients in whom surgical correction was applied is not available
in our study as it origins from an endoscopy department. Though as stated in our result part
and once again underlined in our discussion part (line 4); Overall ERCP achieved leak
resolution in 90.6% of minor
‑leak cases,
Comment
In detecting and treating bile duct injuries, how does ERCP's diagnostic and therapeutic
methodology stack up against MRCP?
Response
Our study does not address this question which is outside the scope of our manuscript. MRCP
according also to guidelines on the topic (ref 6 of our manuscript) is the gold standard for bile
leak diagnosis but cannot be used a therapeutic modality while ERCP can.
Comment
What effect does the lowering of transpapillary pressure gradient caused by ERCP have on
the healing of bile ducts in individuals who have biliary tract injuries?
Answer
When it comes to treating bile leaks, when and how should ERCP be performed?
Response
The question is already answered in our manuscript as our results show that stent
implantation leads to increased healing rates while timely ERCP leads no to increased healing
rates but to bile leak healing acceleration
Comment
What effects do various ERCP procedures, such as sphincterotomy and stent implantation,
have on the rate at which bile leak patients heal?
Response
The question is already answered in our manuscript as our results show that stent
implantation leads to increased healing rates
Comment
When compared to delayed intervention, does early ERCP (within three days) improve
healing outcomes for individuals with bile leaks?
Response
As already included in our analysis ERCP within 7 days does not lead to to increased healing
rates but to bile leak healing acceleration. Since it is reviewers wish we also added in the
results parts analysis about healing rated when comparing ERCP (within three days)
Comment
When treating surgical bile leakage in patients referred from two busy hepato-pancreato-
biliary clinics in Greece, how accurate is ERCP in terms of diagnosis and clinical efficacy?
Response
Clinical efficacy of ERCP is stated already in our study and calculated I our centres to 90.4%.
Concerning ERCPs diagnostic efficacy, such analysis falls outside the scope of our manuscript
as MRCP and not ERCP is the gold standard for bile leak leakage and we did not wish to
question that in our manuscprit
What is the relationship between the identification and treatment of postoperative bile
leakage and the various ERCP referral criteria (drain output thresholds and imaging findings)?
Response
Analysis about drain output thresholds, bile leak injury type according to Strasberg and
existence of abdominal collections found by imaging is already included in our manuscpript
Comment
In treating postoperative bile leaks, what are the procedural results and complication rates of
ERCP carried out by senior endoscopists as opposed to supervised trainees?
Response
This effect cannot be assessed in case of inability of the supervised trainee to conclude the
procedure, this was done by the senior endoscopist and therefore it is impossible to calculate
result and complication rates for senior endoscopists and supervised trainees separately
Comment
What are the safety profiles and procedural results of various stenting and cannulation
approaches in ERCP?
Response
All such data are presented in our result part and commentated on the discussion part
Comment
What effects do various sedation and antiplatelet management techniques have on the
efficacy and safety of ERCP procedures?
Response
Our sedation protocol is the same in all cases the last 25 years so no comparisons can be
made. As far as antiplatelet and anticoagulant medications management is concerned we
have very recently published a large study examining safety and effectiveness of ERCP in
more than 3000 patients submitted to ERCP in our country. Therefore it was deemed
unnecessary to replicate our findings and analysis once more in this study as the aim of this
study was different
Comment
What modifications were made to stenting and anticoagulation techniques in ERCP
operations between the pre-2010 and post-2010 periods
Response
As stated in our method section concerning anticoagulation ‘’ . Antiplatelet and
anticoagulant therapy protocols evolved over time: until 2010, aspi-rin and clopidogrel were
discontinued 5 days prior, and warfarin was managed with bridging therapy; post-2010,
aspirin was generally continued, and DOACs were paused 48–72 hours before ERCP, with INR
monitored to ensure levels below 1.5. Sedation protocols included midazolam with or without
fentanyl initially, evolving to titrated propofol administration based on patient status though
especially concerning sedation protocol a small comment already exist in the study limitation
part which is now enriched (‘’ The 25 year study interval encompassed significant evolution in
ERCP devices, seda-tion practices, and peri procedural care, which may have influenced both
technical success and complication rates though especially concerning sedation protocol our
departments protocol has not changed during the years.’’)
Regarding stenting the first wallstent (metallic stent) was placed in our department on
September 2007 therefore since the first ERCP due to bile leak included in our study was on
January 2002, in only a minority of cases metallic stents were not available (in total 35 cases)
and therefore no comparisons can be made as during the longest period of the study metallic
stents were available. Though a small comment was added in the study limitation part.
Moreover in order to overcome reviewers concerns regarding possible selection bias arising
from the large period of our study we added a small paragraph in our result part comparing
healing rates as also stent usage, time to ERCP and time to bile leak resolution, after dividing
patients related to the time of ERCP execution (before or after the year 2012) – last
paragraph section 3.7.